# Study of the Interseismic Deformation and Locking Depth along the Xidatan–Dongdatan Segment of the East Kunlun Fault Zone, Northeast Qinghai–Tibet Plateau, Based on Sentinel-1 Interferometry

**Shuai Kang, Lingyun Ji \*, Liangyu Zhu, Chuanjin Liu** 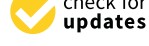 **, Wenting Zhang, Ning Li, Jing Xu and Fengyun Jiang**

The Second Monitoring and Application Center, China Earthquake Administration, 316 Xiying Road, Xi'an 710054, China
\* Correspondence: jilingyun@smac.ac.cn; Tel.: +86-13992856879

**Abstract:** The East Kunlun fault zone (EKFZ), located northeast of the Qinghai–Tibet Plateau, has experienced several strong earthquakes of magnitude seven or above since 1900. It is one of the most active fault systems and is characterized by left-lateral strike-slip. However, the Xidatan–Dongdatan segment (XDS) of the East Kunlun fault zone (EKFZ) has had no earthquakes for many years, and the Kunlun Mountains $M_S$ 8.1 earthquake has a stress loading effect on this segment, so it is widely regarded as a high-risk earthquake gap. To this end, we collected the Sentinel-1 data of the XDS of the EKFZ from July 2014 to July 2019 and obtained the high-precision interseismic deformation field by the Interferometric Synthetic Aperture Radar (InSAR) technique to obtain the slip rate and locking depth of the XDS of the EKFZ, and the seismic potential of the segment was analyzed. The results are as follows: (1) The LOS deformation field of the XDS of the EKFZ was obtained using Sentinel-1 data of ascending and descending orbits, which indicated that the XDS of the EKFZ is dominated by horizontal motion. Combined with the interference results, it is shown that the strike-slip rate dominates the deformation information of the XDS of the EKFZ. The deep strike-slip rate of the fault is about 6 mm/yr, the deep dip-slip rate is about 2 mm/yr, and the slip-deficit rate on the fault surface is about 6 mm/yr; (2) Combined with the spiral dislocation theory model, the slip rate of the XDS to Xiugou Basin of the EKFZ has a gradually increasing trend, with an average slip rate of $9.6 \pm 2.3$ mm/yr and a locking depth of $29 \pm 5$ m; (3) The stress accumulation is about $483 \pm 92$ years in the XDS of the EKFZ, indicating that the cumulative elastic strain energy of the XDS can produce an $M_W$ $7.29 \pm 0.1$ earthquake in the future.

**Keywords:** East Kunlun fault zone; Xidatan–Dongdatan segment; InSAR; interseismic deformation; slip rate; locking depth; seismogenic depth

## 1. Introduction

The East Kunlun fault zone is one of the famous faults and is accompanied by a left-lateral strike-slip structure in mainland China. It stretches for more than a thousand kilometers and traverses the central part of Qinghai Province from east to west. It starts from the west of Whale Lake (the border between Qinghai Province and Xinjiang Province), through Kusai Lake, Xidatan–Dongdatan, Xiugou Basin, Alag Hu, Tuosuo Lake, Xiadawu, and Maqin in the east and extends to the east of Maqu in Gansu Province [1]. Up to now, about 70 paleoseismic events have been discovered in the EKFZ in Qinghai Province, and these earthquakes are approximately evenly distributed in various segments between Katabanfeng and Maqu, forming a vast earthquake surface fracture zone and stretching nearly 1000 km. Through the analysis of the paleoseismic catalog, it is concluded that the number of historical earthquakes in the EKFZ accounts for about 3/4 of the entire Bayanhar Mountains seismic zone, and it is the central active region of the Bayanhar

Mountains seismic zone. The frequency of the EKFZ seismic activity from high to low is as follows: the west segment, the east segment, the middle west segment, and the middle east segment [2–4].

Since 1900, the EKFZ has experienced several strong earthquakes of magnitude seven or above, such as the 1937 Huashixia $M_S$ 7.5 earthquake, the 1963 Dulan $M_S$ 7.0 earthquake, the 1973 Mani $M_S$ 7.3 earthquake, the 1997 Mani $M_S$ 7.5 earthquake, and the 2001 Kunlun Mountains $M_S$ 8.1 earthquake. These earthquakes are mainly concentrated in the western and middle eastern segments of the EKFZ [5,6]. Among them, the 2001 Kunlun Mountains $M_S$ 8.1 earthquake had a stress loading effect on the Xidatan–Dongdatan segment (XDS) of the EKFZ [7–12]. Therefore, the XDS of the EKFZ is worthy of attention for future strong earthquakes. Scholars have also verified this view through paleoseismic identification, earthquake repetition interval determination, and model probability calculation (e.g., [3,7–10]). Based on a variety of basic methods such as seismic geology, geodesy, and geophysics, the Working Group of Seismic Situation in Mainland China gives the determination results of key earthquake risk areas in China from 2021–2030, among which the XDS of EKFZ is one of the key risk areas (red dashed circle in Figure 1a; [11,12]). At present, the study of the tectonic movement characteristics of the EKFZ mainly focuses on the long-term slip rate on the eastern and western members, such as, from west to east, the slip rates of the EKFZ are 10.0–2.0 mm/yr [13] and 14.8–9.4 mm/yr [14]; the slip rate of Maqin since the late Pleistocene is 12.5 mm/yr [15]; from late Pleistocene to Holocene, the slip rate of the Maqu segment is 10.15–2.0 mm/yr, and it is speculated that the slip rate of the Maqu segment shows the gradient attenuation [16–19]. On the west segment of the EKFZ, the slip rate is 10–12 mm/yr [20], and in the Mani earthquake of the west segment of EKFZ, the interseismic slip rate is 10.2 mm/yr [21]. However, there is less research on the XDS, which is a large earthquake gap with a higher future earthquake risk.

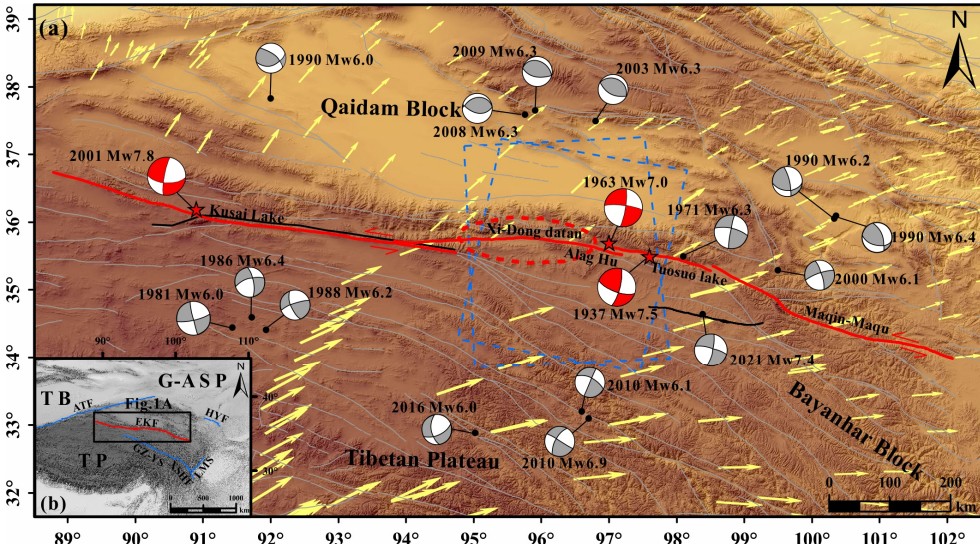

**Figure 1.** Seismotectonic background map of the East Kunlun Fault Zone (EKFZ). Faults are modified from [22] and are superimposed on the Digital Elevation Models (DEMs) from the Shuttle Radar Topography Mission (STRM). (**a**) The red dashed circle is the middle and long danger zone from 201–2030. The bold red line represents the EKFZ. The red five-pointed stars represent the historical earthquakes of the EKFZ. The dashed blue box indicates the InSAR data coverage. The black lines represent the rough surface rupture of the historical earthquakes [23,24]. The black dots are the historical earthquakes surrounding the EKFZ. The gray lines represent secondary faults around the study area. The yellow arrows represent GPS velocity fields at the regional. (**b**) Macroscopic distribution of the study area. T B is the Tarim Basin. G-A S P is the Gobi-Ala Shan Platform. T P is the Tibetan Plateau. The blue lines show the main faults around the EKFZ. Altyn Tagh Fault (ATF), Haiyuan Fault (HYF), Ganzi–Yushu Fault (GZ-YS), Xianshuihe Fault (XSHF), Longmenshan Fault (LMS).

The XDS of the EKFZ is located between the 2001 Kunlun Mountains $M_S$ 8.1 earthquake and the 1963 Dulan $M_S$ 7.0 earthquake, and it is a seismic gap in the EKFZ and an earthquake risk region in Mainland China (Figure 1a). It is of great practical significance to study the current locking degree and slip deficit characteristics of the XDS for further understanding of earthquake risk in this region and more scientific and practical earthquake prevention and disaster reduction. In recent years, with the accumulation of InSAR data and the continuous updating of data processing technology, not only can we obtain the crustal deformation field during interseismic periods in fault zones but also timely and effectively obtain large-range high-precision deformation and strain accumulation characteristics of fault by InSAR technology (e.g., [19,25–30]). As shown in Figure 1a, the altitude of the study area is relatively high, resulting in less GNSS measurement data, but SAR satellite data can comprehensively cover it. From July 2014 to July 2019, there are data on both the ascending and descending orbits on the XDS of the EKFZ, which can ensure the integrity of the data and no earthquakes occurred during this period, which can ensure the reliability of the results of the interseismic deformation field of the XDS of the EKFZ. Therefore, we collected the Sentinel-1 data of the XDS of EKFZ from July 2014 to July 2019 to study the kinematic characteristics (such as the locking depth and the slip rate) of the XDS combined with InSAR technology. Based on this, the sectional activity of the XDS of the EKFZ and the difference in tectonic movement among each segment are discussed, and the seismic potential or magnitude in this area is analyzed.

## 2. InSAR Data and Interseismic Velocity Field

### 2.1. InSAR Data

Interferometric Synthetic Aperture Radar (InSAR) was first proposed by L.C. Graham [31]. It is a three-dimensional imaging concept and is used as a spatial geodetic technology developed in recent decades. Unlike traditional geodesy techniques (GPS, Leveling, etc.), InSAR technology can measure crustal deformation fields with high spatial resolution [31–35]. At present, SAR satellite data are rich, have high precision, are not constrained by time and climate, etc., and are widely used in monitoring geological disasters, earthquake disasters, volcanic activities, city subsidence, and railway subsidence (e.g., [36–43]). This paper uses data from the European Space Agency (ESA) Sentinel-1 satellite. Compared with other satellites, Sentinel-1 has the advantages of a short revisit period and high orbital accuracy [33]. Since the launch of the satellite, Sentinel-1 data have been widely used in the study of coseismic deformation, such as the 2014 Napa California $M_W$ 6.1 earthquake [38,44,45], the 2015 Nepal $M_S$ 8.1 earthquake [39,46], and the 2017 Jiuzhaigou $M_S$ 7.0 earthquake [40,47]. With the accumulation of SAR satellite data over the years and the continuous improvement of InSAR data processing technology, InSAR technology should also obtain an extensive range and high-resolution fault tectonic deformation field (e.g., [19,25,27–30,48,49]).

### 2.2. InSAR Data Processing Method

In this paper, InSAR data are processed using the GAMMA commercial software platform [50]. The interseismic deformation field was mainly obtained from the Sentinel-1 SAR satellite data by using the Stacking method. The method was proposed by Wright et al. [51] in 2001. It is a sequential InSAR algorithm that utilizes linear stacking of multiple unwrapped differential interferometry phase graphs to minimize atmospheric errors and improve the accuracy of the deformation rate [51–53]. The principle of the Stacking method assumes that in an independent interferogram, the error phase of atmospheric disturbance is random, and the deformation rate is linear. Then, the average deformation rate in the superposition time range of the study area can be obtained by calculating the average superposition of the phases unwinding corresponding to multiple independent interferograms [51–56]. The specific data processing flow is shown in Figure 2.

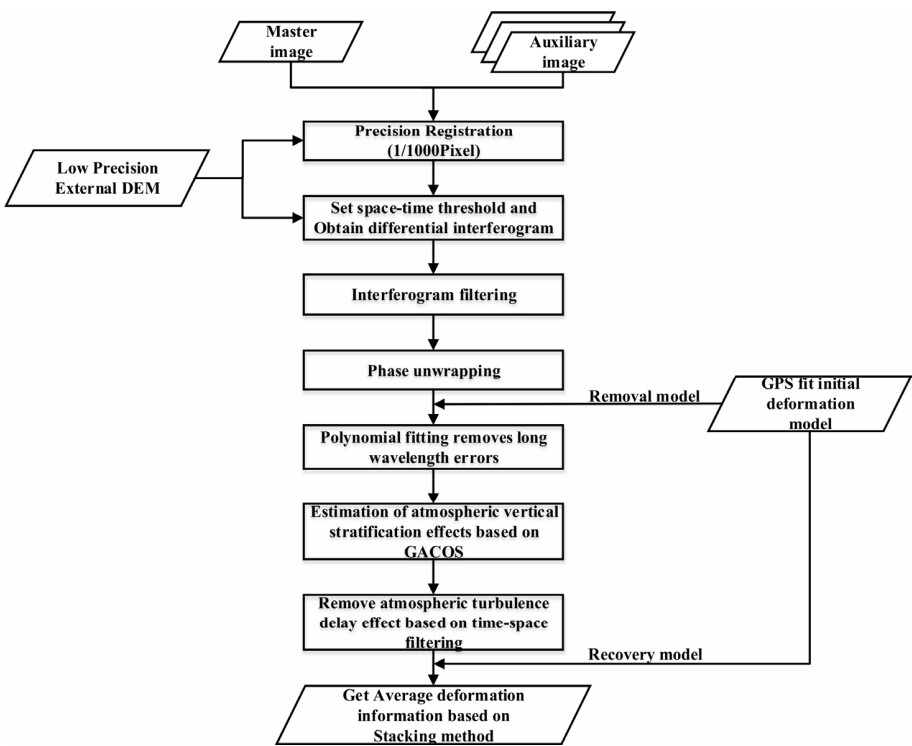

**Figure 2.** Sequential InSAR algorithm processing flow [55,57].

In this study, a total of 136 Sentinel-1 SAR images covering the XDS of the EKFZ were obtained from July 2014 to July 2019, with 64 ascending and 72 descending orbit images. The spatial range of these data was centered on XDS, 330 km in the NS directions and 280 km in the EW directions (Figure 1a; Table 1). We set the multi-look factor of the interferogram to 100:30, and the corresponding pixel size was about 420 m × 420 m, from which pixels with a coherence degree of about 0.2 were selected for phase unwrapping. Interferograms spanning a short time interval tend to have higher coherence. However, they make small tectonic signals, such as those across the XDS of the EKFZ, more difficult to detect. To minimize geometric decorrelation and errors due to the topography and maximize the deformation signal in the interferogram [25,51–53], we selected pairs of images by constraining the perpendicular and temporal baselines simultaneously. So, to improve the signal-to-noise ratio of interseismic deformation, we selected interferograms based on two criteria (Table 2; Figure 3). (1) Image pairs with temporal baselines of different Days were employed to maximize the deformation signal in the interferograms. (2) Due to the topography of the XDS of the EKFZ being very undulating, spatial baselines (separation between orbits) of less than 10 m, 100 m, and 200 m were used to minimize topography error influences, respectively. As can be seen from Table 2, the temporal and perpendicular baselines are coordinating relations and should be met simultaneously. (1) When the temporal baseline is less than 90 days, the perpendicular baselines must be less than 10 m; (2) When the temporal baselines are between 275 and 455 days, the perpendicular baselines must be less than 100 m; (3) When the temporal baselines between $n + 275$ and $n + 455$ days ($n$ is an integer multiple of 365), the perpendicular baselines must be less than 200 m.

**Table 1.** Sentinel-1 SAR image and interferogram count.

| Flight Direction | Track Number | Number of Images | Number of Selected Image Number | Number of Interferograms | Number of Selected Interferograms |
|---|---|---|---|---|---|
| A * | 172 | 64 | 63 | 662 | 165 |
| D * | 004 | 72 | 41 | 974 | 304 |

* A is the ascending orbit, and D is the descending orbit.

**Table 2.** Constraint Criteria of Temporal and Perpendicular Baselines.

| No. | Temporal Baseline (Days) $\Delta T$ | Perpendicular Baseline (Meter) $\Delta P$ |
|---|---|---|
| 1 | $\Delta T < 90$ Days | $\Delta P \leq 10$ m |
| 2 | 275 Days $< \Delta T < 455$ Days | $\Delta P \leq 100$ m |
| 3 | $n + 275$ Days $< \Delta T < n + 455$ Days * | $\Delta P < 200$ m |

* $n$ is an integer multiple of 365.

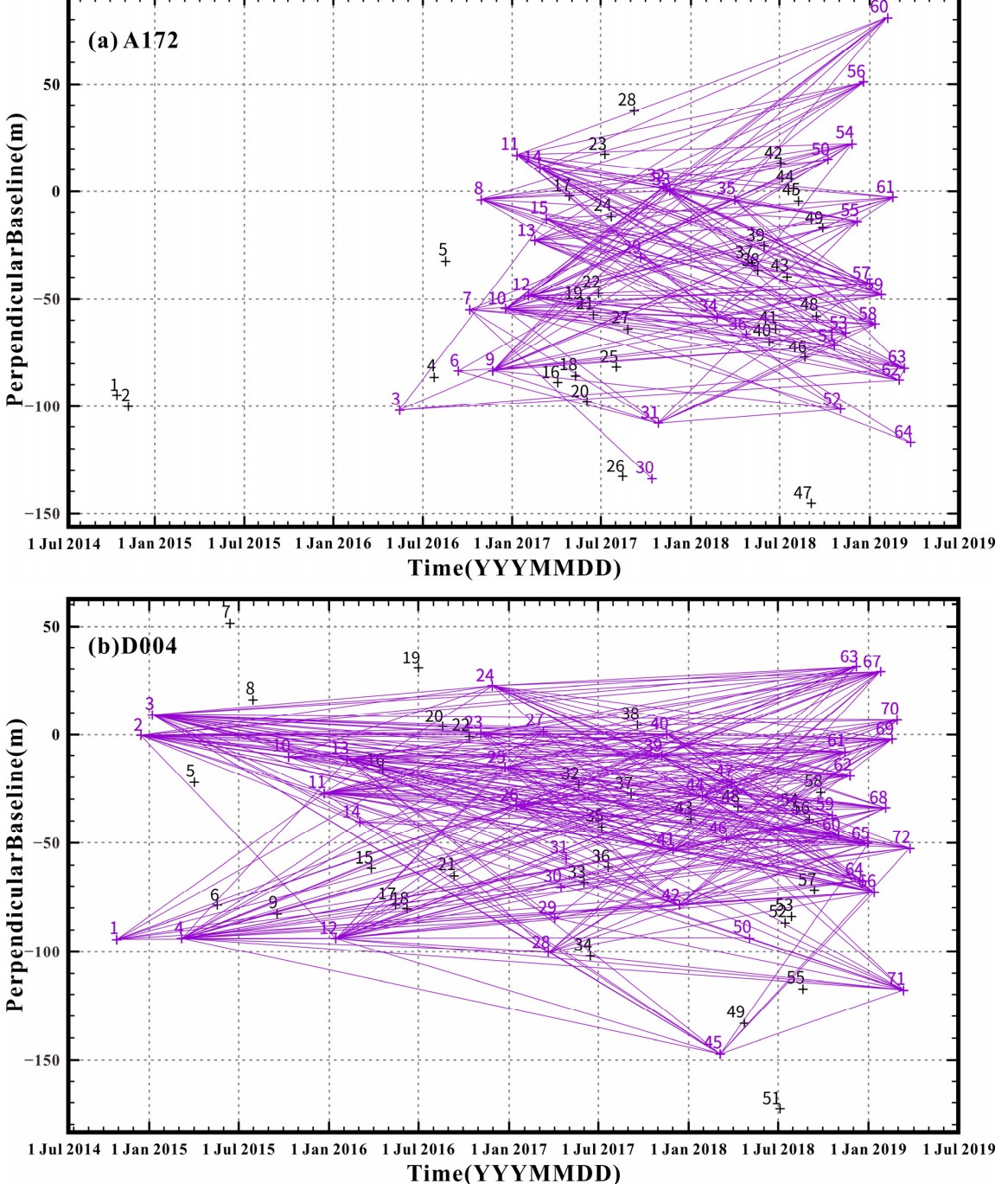

**Figure 3.** Distribution of temporal and perpendicular baselines of the interferogram. The images included in the inversion calculations are represented by the purple + labels. The images not included in the inversion calculations are represented by the black + labels. The interferograms are represented by purple lines. (**a**) The ascending orbit 172; (**b**) The descending orbit 004.

### 2.3. InSAR Inversion Interseismic Velocity Field

The inverse slip dislocation theory is a commonly used method to describe slip rate distribution in the interseismic period [58,59]. This theory assumes that the movement of all points inside the fault is determined by the rotation of the fault and the deformation inside the fault, and then the surface deformation caused by the slip deficit is determined by the blocking depth at the fault surface. Therefore, based on the inverse slip dislocation theory, various models (such as the angular dislocation model, rectangular dislocation model, elasticity model, viscoelasticity model, and layered viscoelasticity model) developed to simulate the observed velocities in the interseismic period by researchers and implemented them with code and software, such as TDEFMPDE code [60], Blocks code [61,62], and the software of viscoelastic model layered [63,64]. Since the viscoelastic properties of the crustal medium have no significant influence on the observed interseismic velocity [19,64], the slip rate of the XDS of the EKFZ in the interseismic period was estimated using the elastic medium.

As shown in Figure 1a, multiple branch faults in the XDS of the EKFZ are distributed, and the tectonic background is complex [19,65], so the far-field loading rate [66] or geological slip rate [67] could determine the long-time slip rate of the XDS of the EKFZ. On this basis, combined with the inverse slip dislocation theory, it can be seen that the locking degree of the fault interseismic period is a quotient of the interseismic slip-deficit rate and the interseismic long-time slip rate [19,59]. Therefore, we assume that the interseismic deformation observed by InSAR expresses the loading rate of the far field [61,62,67–71]. A reliable and more informative constrained least square method is used to invert the interseismic slip distribution of the XDS of EKFZ [19,72]. Then, the degree of current fault surface occlusion is estimated.

## 3. Study of the Interseismic Velocity Field in the XDS

### 3.1. InSAR Interseismic Velocity Field Analysis

Sentinel-1 SAR data processing mainly involves several critical problems: (1) In image precision registration, to ensure the accuracy of image registration, orbit information, external Digital Elevation Model (DEM), and intensity map should first be used for initial registration, and then enhanced spectral diversity should be used for azimuth registration to achieve fine registration [54,73]; (2) When generating differential interferogram, use the 30 m spatial resolution SRTM DEM released by NASA to simulate and eliminate terrain phase; (3) Considering that dense vegetation, snow and ice cover, climate variability, and other factors in the study area lead to poor coherence of the interferogram, it is necessary to conduct multi-view processing and filtering processing on the interferogram. In the interferogram process, two multi-look processes are adopted: the first multi-look factor was set to 10:3, and the second multi-look factor was set to 100:30. After multi-view processing, the Goldstein filter was used to improve the coherence further [74]; (4) Based on prior fault information, the initial deformation model was constructed to estimate and remove orbit errors; (5) The Generic Atmospheric Correction Online Service (GACOS) was used to estimate the effects of atmospheric vertical stratification, and time-domain high-pass and spatial-domain low-pass filtering were used to estimate the effects of atmospheric turbulence delay [75,76]. Finally, the high accuracy and high spatial resolution mean deformation rate field was obtained in the study area based on the mature temporal InSAR technology Stacking (Figure 4).

As shown in Figure 4, the ascending orbit deformation is opposite to the descending orbit deformation direction, indicating that horizontal movement is the primary movement in this area. In the vicinity of the XDS of the EKFZ, there is the maximum difference in the InSAR deformation field of ascending and descending orbits, and they accord with the movement characteristics of a left-lateral strike-slip fault, indicating that the XDS of the EKFZ mainly influences the crustal deformation in this area. In Figure 4a, the InSAR deformation field in the LOS direction of the A172 has a significant velocity difference of about 2–5 mm/yr along the fault and its extension, and the velocity step is mainly near the

XDS of the EKFZ. There is no significant large-scale difference in other faults. Figure 4b shows that the InSAR deformation field in the LOS direction of the D004 shows this feature and a minor velocity difference of about 1–4 mm/yr near the fault. Therefore, the overall deformation characteristics of the XDS of the EKFZ are small near-field movement and large far-field movement, which is a typical feature of strike-slip fault [19,58].

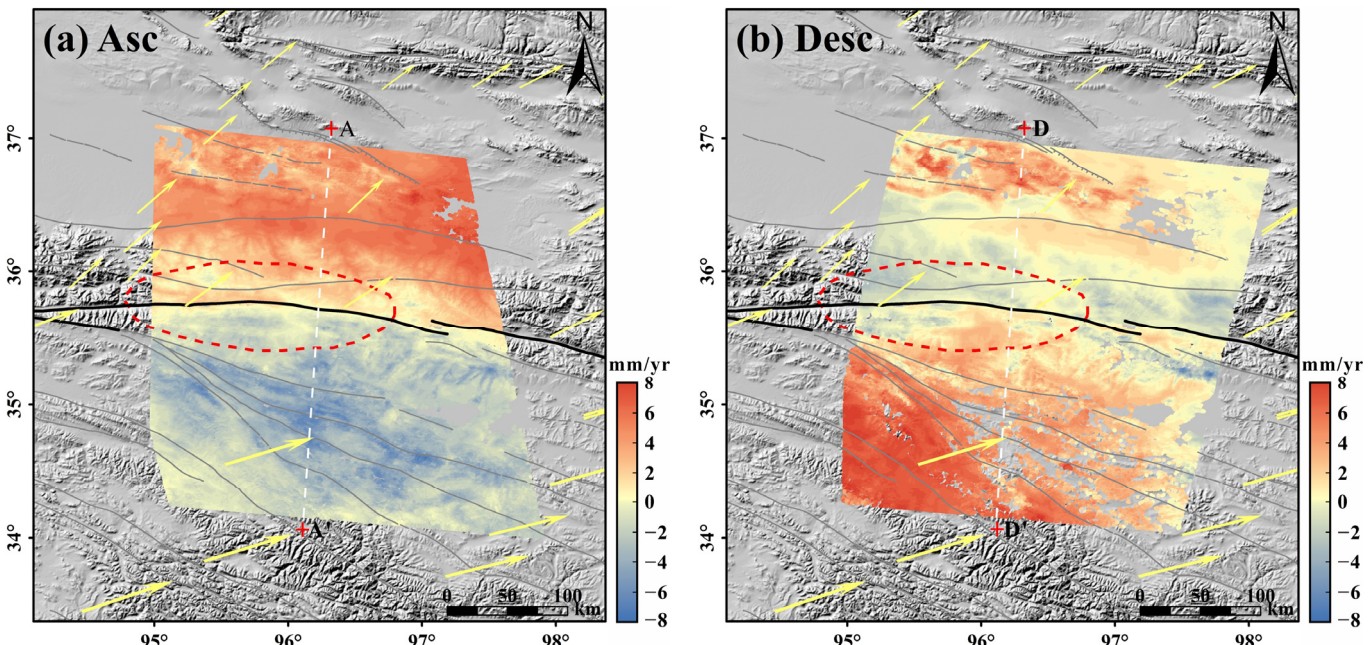

**Figure 4.** InSAR interseismic velocity field of the XDS. The red dotted circle represents the danger zone (same as in Figure 1). The black line represents the main fault of the EKFZ, The yellow arrows represent GPS velocity fields at the regional, and the gray lines represent the faults around the EKFZ [22]. (**a**) The ascending orbit 172, AA': the profile line of the ascending orbit. (**b**) The descending orbit 004, DD': the profile line of the descending orbit.

### 3.2. Accuracy Evaluation of InSAR Interseismic Velocity Field

There are two main methods to appraise the precision of the InSAR interseismic deformation results. The first method compares it with GPS results; the second method calculates the standard deviation (SD) of the velocity differences in the overlapping area of adjacent independent orbits [19,30,54,77,78]. This paper used the image data of the ascending and the descending, so the accuracy was by GPS results. We extracted the profile line of the ascending orbit (AA') (Figure 4a) and the descending orbit (DD') (Figure 4b), respectively, to compare the GPS and InSAR results. The profile line position is shown in Figure 4. The profile line length was 150 km on each side of the fault. GPS data were extracted within 120 km on each side of the profile line. InSAR data were extracted within a 50 km width on each side of the profile line. Since InSAR data and GPS data respectively represent deformation rates in different directions, and the activity of the EKFZ is a left-lateral strike-slip, we can ignore the vertical motion component of the EKFZ [19,79]. Therefore, before data extraction, we used the mathematical Formula (1) to convert the GPS-observed data into the deformation rate in the LOS direction using the flight azimuth and radar incidence angle [56,80].

$$D_{LOS} = (-\sin\theta\cos\alpha) \times D_{EW} + \sin\theta\sin\alpha \times D_{NS} \tag{1}$$

where $D_{LOS}$ is the deformation rate in the LOS direction of radar, $D_{EW}$ and $D_{NS}$ are the horizontal east–west deformation rate and north–south deformation rate of GPS, $\alpha$ is the flight azimuth angle, and $\theta$ is the radar incidence angle. Secondly, to correct the offset between the GPS data and the reference datum of the InSAR data, the average value corrects

the extracted GPS rate. It is compared with the deformation rate in the InSAR LOS direction. The profile of GPS and InSAR deformation rate is shown in Figure 5. From the perspective of visual effect, the two independent data sets of GPS and InSAR have a high degree of agreement. The independent horizontal GPS measurements agree well with the InSAR measurements, which not only supports the hypothesis of minimum vertical deformation but also confirms that the EKFZ movement is mainly horizontal [19,79]. Finally, the SD is 1.87 mm/yr between the InSAR deformation rate of the ascending LOS and GPS, and the SD is 2.45 mm/yr between the InSAR deformation rate of the descending LOS and GPS. Therefore, it can be concluded that the measurement accuracy of the InSAR interseismic velocity field is better than 3 mm/yr.

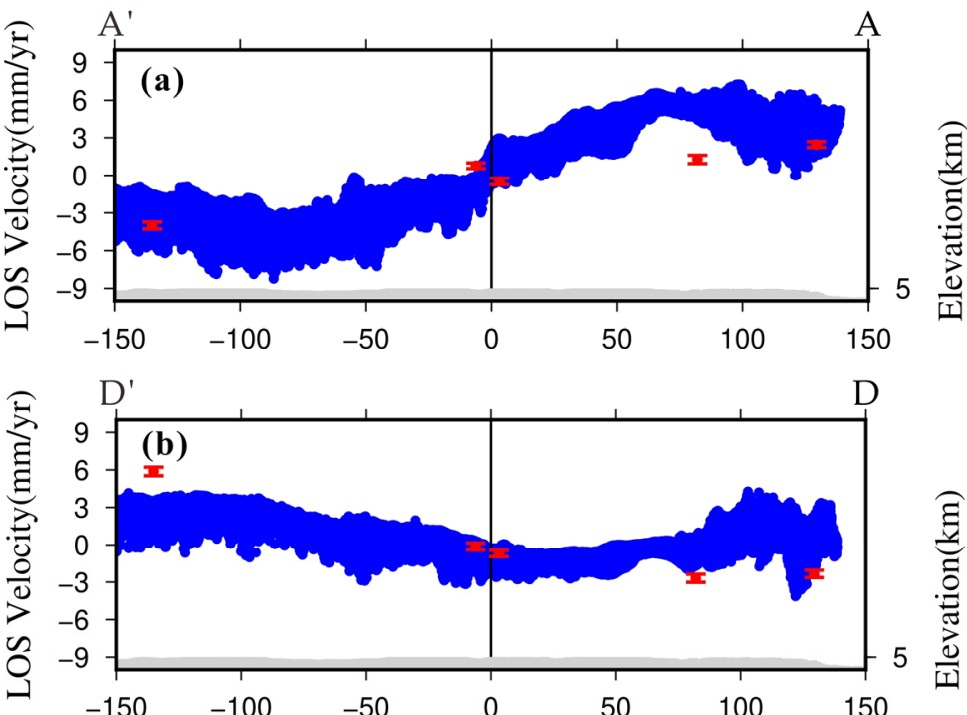

**Figure 5.** Accuracy evaluation showing the profile of GPS and InSAR deformation rates. The GPS measuring results are represented by red diamonds with error bars. The InSAR measuring results are represented by blue circles. The locations of the EKFZ are represented by the black vertical lines. The topographic information is represented by the gray areas. (**a**) AA′ profile in the ascending orbit and (**b**) DD′ profile in the descending orbit.

## 4. Interseismic Deformation Modeling Based on InSAR

### 4.1. Parameters and Smoothing Factors of the Fault Geometry Model

The geometric model parameters of the fault should be determined before the interseismic deformation model based on the InSAR-derived model. The following reasons should be considered before establishing the geometric models of the XDS of the EKFZ. (1) From Figure 4, we can see that the EKFZ mainly influences all the deformation information in the research area, so we do not need to consider the effects of other faults during the inversion process; (2) According to the previous studies, the dip of the XDS of EKFZ ranges from 70° to 90° [19,81]. We performed some search calculations to determine the fault dip angle. After varying the fault dip angle intervals and calculations several times, the results are shown in Figure 6a, and if the fault dip angle of the XDS of the EKFZ was set to 90°, the fitting residual was the smallest. Therefore, the fault dip angle of the XDS of the EKFZ was set to 90° when the fault model was established (Figure 6b); and (3) According to the inverse slip dislocation theory, the fault planes can be divided into upper and lower parts [71,82]. The upper part is responsible for the near-field deformation signal and is assumed to be the seismogenic fault. Its depth is determined to be 20 km by the source depth of the

repositioned earthquake. The lower part is responsible for the far-field deformation signal and is assumed to be an elastic layer, and its depth is set to 80 km according to the depth of the lower boundary of the lithosphere [83]. In addition, we tested the slip rate distribution at depths of 20 km, 40 km, 60 km, and 80 km, respectively, and concluded that the slip rate distribution varied little when the section depth was about 40 km.

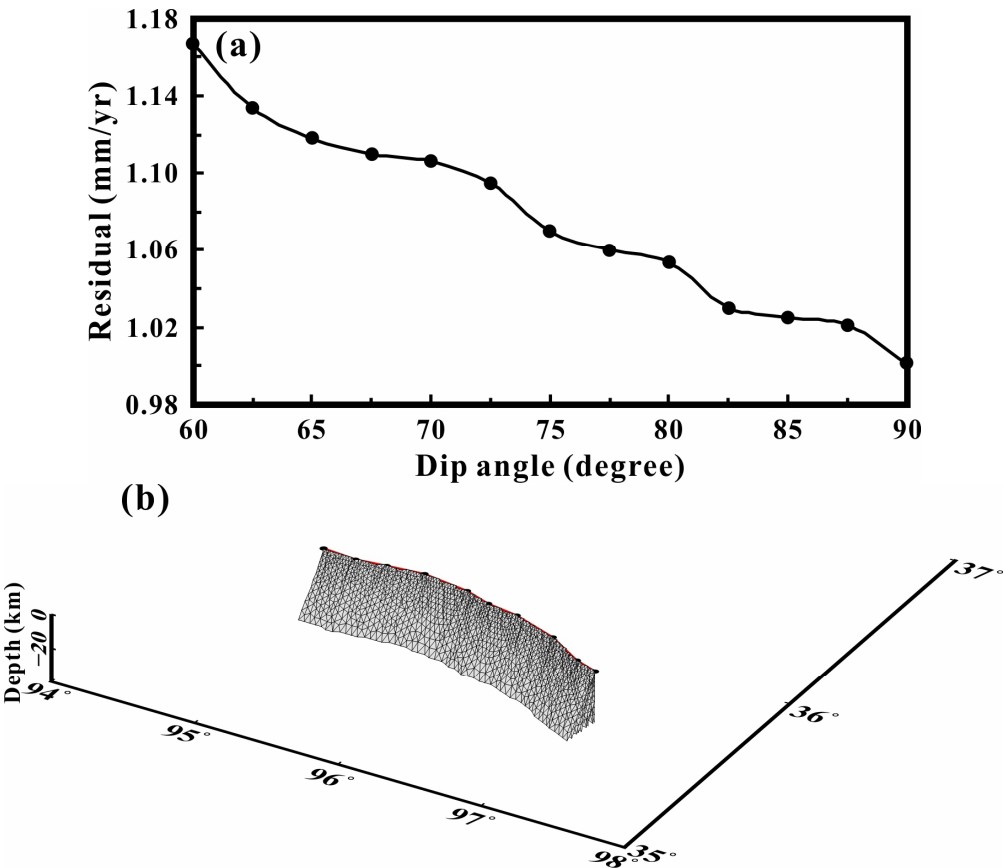

**Figure 6.** Results of the fault dip angle residual search and effect diagram of fault modeling. (**a**) Variation of the residual with different fault dip angles. (**b**) Variation of fault geometry with different fault dip angles.

### 4.2. Modeling Effects and Locking Degree Distribution in the XDS

The angular dislocation model can not only depict complex fault planes well but also avoid tearing caused by strike-slip direction deformation [64,68,84]. Therefore, the interseismic slip rate distribution in the XDS of the EKFZ was inverted with the InSAR average velocity field as a constraint (Figure 4). Figure 7 shows the interferogram fitting results, in which Figure 7a,d show the InSAR observation results of the ascending and descending orbits; Figure 7b,e show the interference results of the ascending and descending orbits; Figure 7c,f show the residual results of observation results and interference results; Figure 7g,h show the residual histograms of Figure 7c,f respectively; and Figure 7i shows the curve of model roughness and fitting residual L, with the optimal smoothing factor of 0.5. The overall fitting is reasonable, combined with the deformation characteristics of the region and the deformation characteristics of the near field of the fault. However, because some artifact signals in the local far-field signals were not entirely removed in the preprocessing stage and could not be fitted, they were deducted. Even if they were true deformation signals, they were too far from the XDS of the EKFZ to have anything to do with the movement of the XDS of the EKFZ.

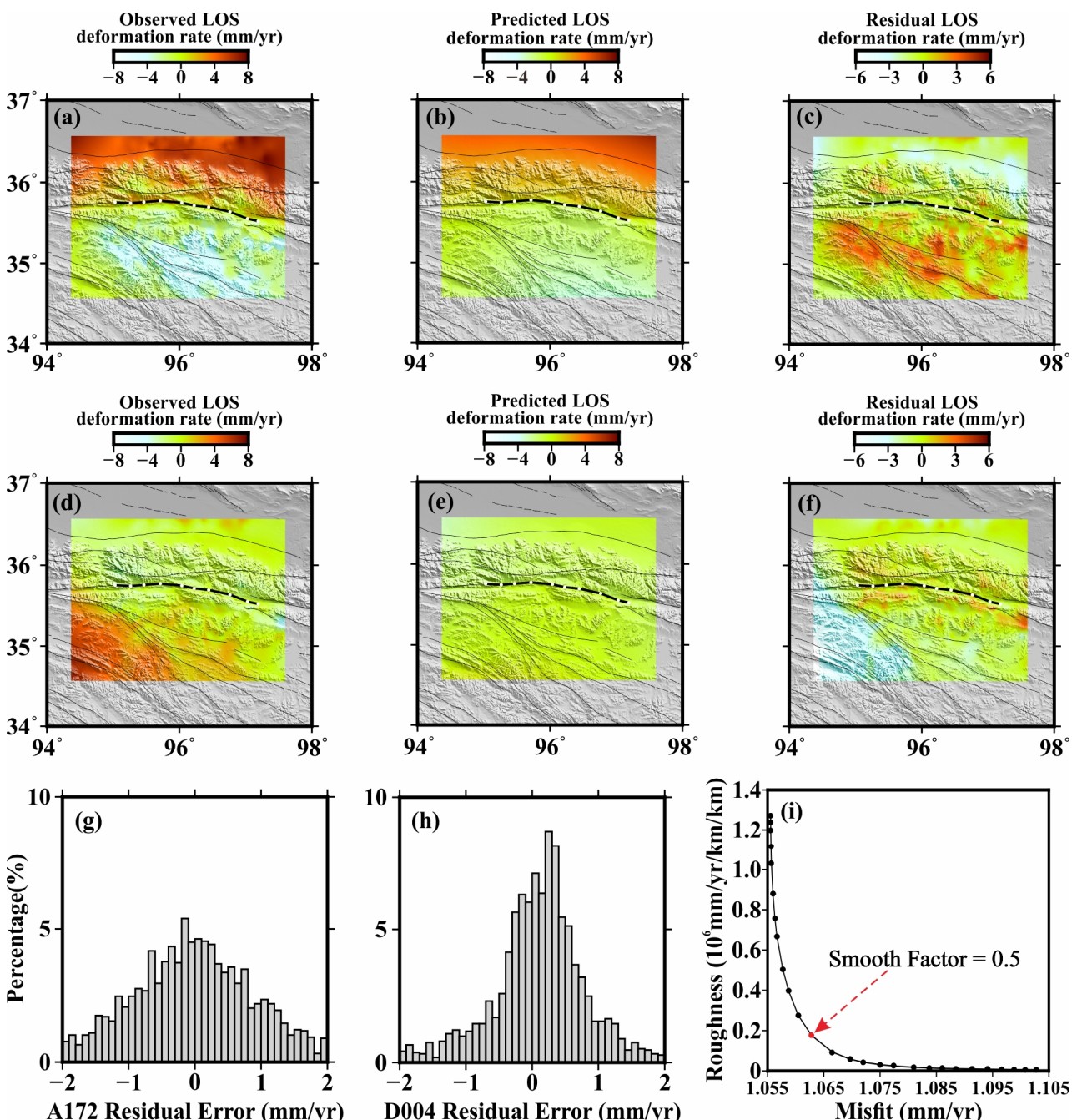

**Figure 7.** Modeling results and error statistics. The black line represents the main fault of the XDS of the EKFZ. (**a**,**d**) The InSAR observation results of the ascending and descending orbits. (**b**,**e**) The interference-fitting results of the ascending and descending orbits. (**c**,**f**) The residuals of the two results. (**g**,**h**) The residual histograms of (**c**,**f**). (**i**) The L-curve of model roughness and fitted residual, and the red dot represents the optimal smoothing factor.

Figure 8 shows the distribution of the slip rates on the fault plane, and the fault is in motion in the depth range. In Figure 8a, the deformation information of XDS of the EKFZ is dominated by the strike-slip rate. The deep strike-slip rate of the fault is about 6 mm/yr, and the maximum locked depth is about 30 km. In Figure 8b, the distribution of the dip-slip rate is limited to 2 mm/yr in the fault deep. In Figure 8c, the distribution of the slip loss is about 6 mm/yr on the fault surface.

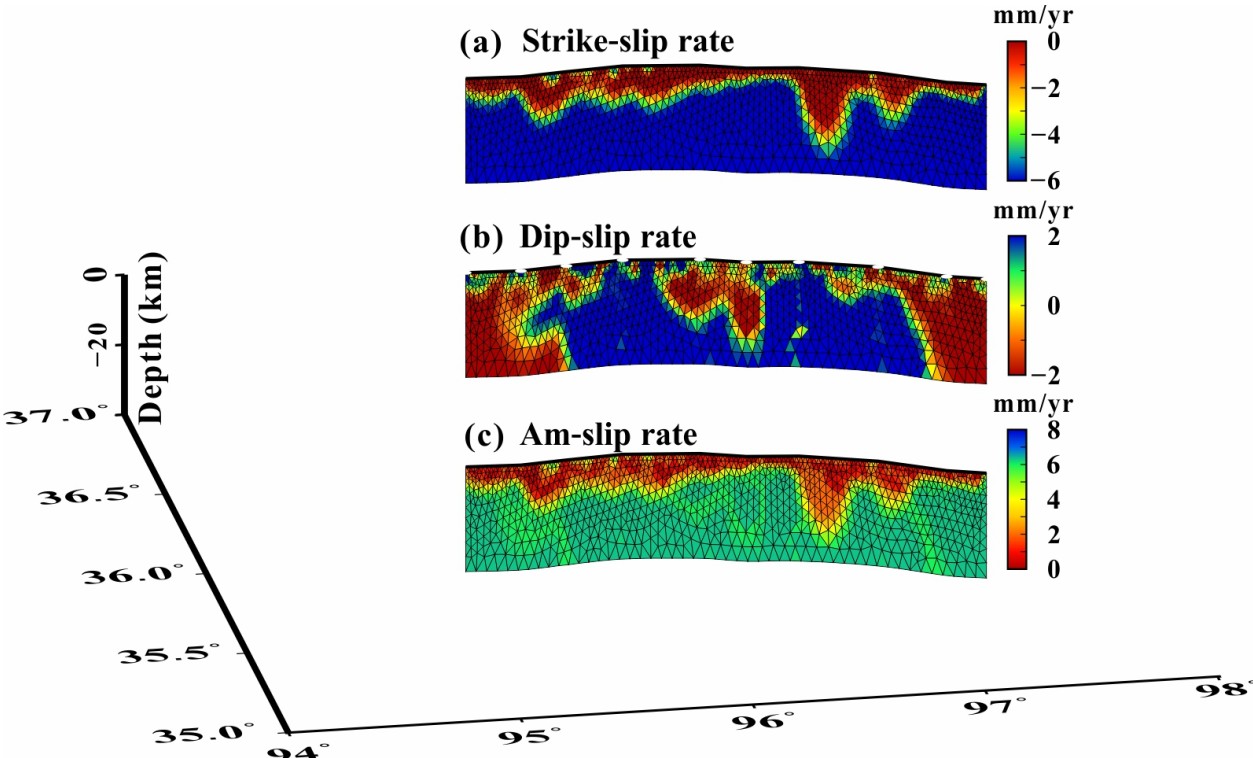

**Figure 8.** Interseismic slip distribution of the fault plane. (**a**) Strike-slip rate, in which negative values indicate left-lateral strike-slip motion to the east. (**b**) Dip-slip rate, in which the red (negative) values represent downward motion. White (positive) values indicate upward motion. (**c**) Am-slip rate represents the distribution of the slip-deficit rate on the fault plane.

## 5. Discussion

### 5.1. Present-Day Strain Distribution around the XDS of the EKFZ

It can be seen from Figure 4 that the InSAR deformation field captured the deformation gradient of the XDS of the EKFZ. However, there are other large local deformation gradients in the deformation field, which may be related to glacier, landslide, seasonal frozen soil, or other non-tectonic factors [85]. Therefore, the three-dimensional deformation field of the XDS of the EKFZ was constructed by integrating the previously published GNSS observations [86,87] combined with the InSAR deformation observations obtained from the ascending and descending orbits (Figure 9a–c). Figure 9a shows the E-W velocity component in the study area, and a velocity gradient was observed along the XDS of the EKFZ, indicating that a shear strain rate concentration is located in this region. Figure 9b shows the N-S velocity component in the study area, and there is no obvious velocity gradient along the XDS of the EKFZ. Figure 9c shows the velocity component in the vertical direction, and there is no obvious velocity gradient along the XDS of the EKFZ.

The strain rate field is one of the important constraints on tectonic deformation and allows for assessing seismic risks [19,42,88–90]. In this study area, we obtained the second strain rate invariant of the XDS of the EKFZ based on the three-dimensional deformation field (Figure 9d). As can be seen from Figure 9d, the accumulation of strain is mainly concentrated in the southeast direction of the Kusai Lake segment of EKFZ (F1-1), the XDS of the EKFZ (F1-2), and Tuosuo Lake segment (F1-3), with the accumulation amount of about 80 nano-strain, and other regions are diffused. It can be inferred that the EKFZ dominates the strain model in the Qinghai–Tibet area and that the 2001 Kunlun Mountains $M_S$ 8.1 earthquake has a loading effect on the strain accumulation of F1-1 and extends to the southeast direction.

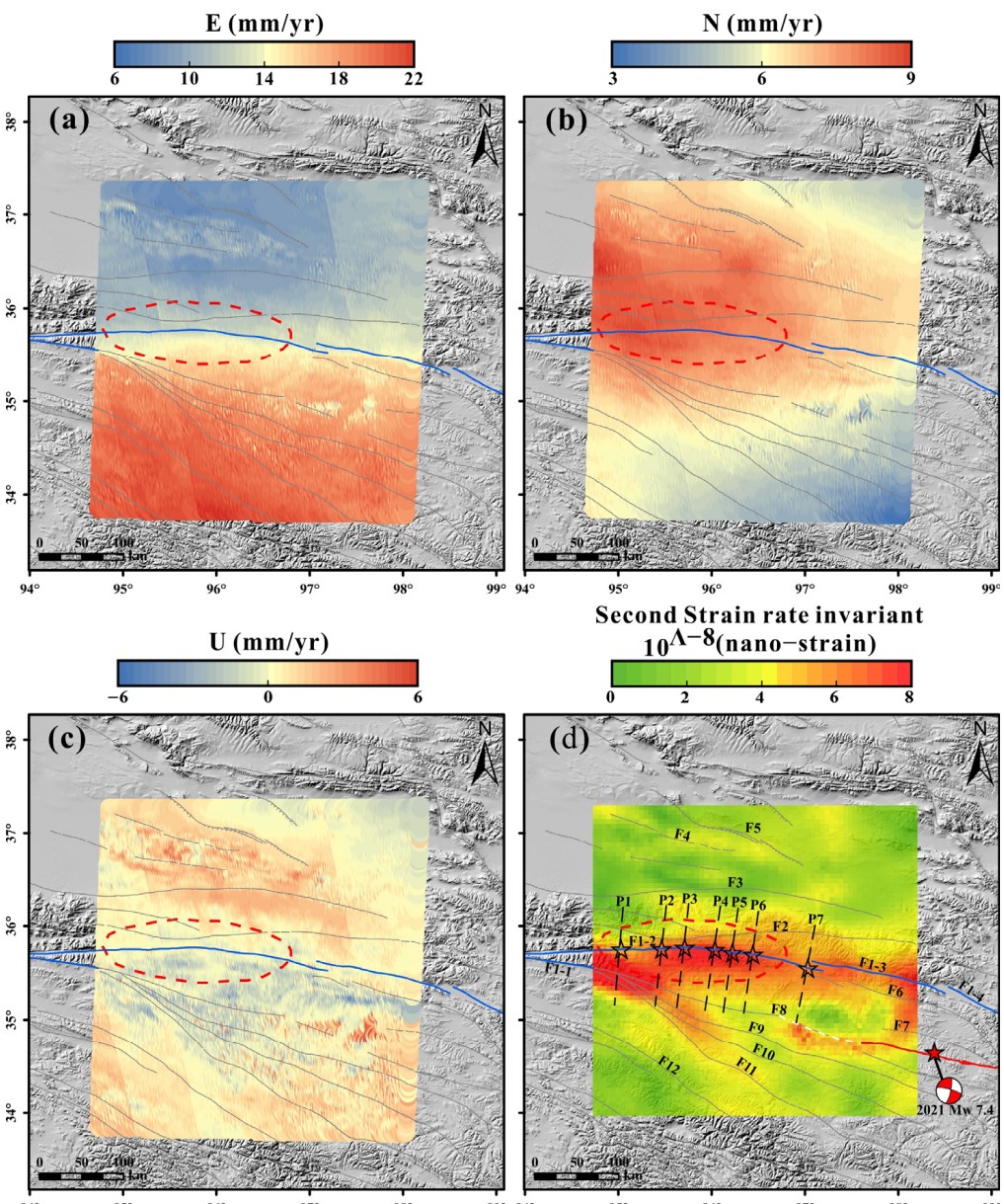

**Figure 9.** The three-dimensional deformation field and second invariants of the strain rates of the XDS. The blue line is the main fault of the EKFZ, and the gray line represents faults (same as in Figure 4) [22]. The red dotted circle represents the danger zone (same as in Figure 1). (**a**) Velocity in the eastern direction. (**b**) Velocity in the northern direction. (**c**) Velocity in the vertical direction. (**d**) Second invariants of the strain rates (The black five-pointed star is the fault position obtained by the inversion of spiral dislocation. The black dashed line is the position of the fit profile. The red five-pointed star is the 2021 Maduo $M_W$ 7.4 earthquake. The white dotted line is presumed to be the extension direction of the Jiangcuo fault. The red line is the Maduo surface rupture [24]). (F1-1: Kusai Lake segment of EKFZ, F1-2: the XDS of EKFZ, F1-3: Tuosuo Lake segment of EKFZ, F1-4: Maqin–Maqu segment of EKFZ, F2: Middle Kunlun fault, F3: Qaidam south margin fault, F4: Qaidam north central fault, F5: Qaidam north margin fault, F6: Dagou–Changmahe fault, F7: Maduo–Gande fault, F8: Jiangcuo fault, F9: Dari fault, F10: Bayanqala fault, F11: Wudaoliang–Changsha Gongma fault, F12: Wudaoliang–Qumalai fault).

In the strain rate diagram (Figure 9d), we noticed that the strain variables east of the XDS of the EKFZ are not concentrated on a single fault, such as the Middle Kunlun fault (F2), Dagou–Changmahe fault (F6), and Maduo-Gande fault (F7). This dense strain rate

field also accords with the diffuse distribution, and it can be seen that the strain rate is absorbed not only by the fault systems but also by the deformation of the small blocks separated by faults. The faults north of the EKFZ include the Qaidam south margin fault (F3), Qaidam north central fault (F4), and Qaidam north margin fault (F5), and they are collectively referred to as the Qaidam fault. The strain rate is not significantly at the Qaidam fault but cannot be ignored. The faults south of the EKFZ include Jiangcuo fault (F8), Dari fault (F9), Bayanqala fault (F10), Wudaoliang–Changsha Gongma fault (F11), and Wudaoliang–Qumalai fault (F12). It is noteworthy that the strain rates of Jiangcuo, Dari, Bayanqala, and Wudaoliang–Changsha Gongma faults are significant, which may be mainly caused by the 2001 Kunlun Mountains $M_S$ 8.1 earthquake. From a regional perspective (Figure 1a), the Jiangcuo fault belongs to the secondary fault on the south side of the EKFZ, and its westward extension could be connected with the 2001 Kunlun Mountains $M_S$ 8.1 earthquake, so the predecessor also referred to the two as Kunlun Mountain Pass– Jiangcuo fault [91]. The Maduo earthquake occurred in 2021, and field research showed that the earthquake produced a coseismic surface rupture zone with the characteristics of a left-lateral strike-slip in a field of about 151 km (Red line in Figure 9d). Based on this, it is inferred that the Jiangcuo fault extends along the southeast direction (white dotted line in Figure 9d) and is connected with the surface rupture generated by the 2021 Maduo $M_W$ 7.4 earthquake. Further, it confirms that the 2021 Maduo $M_W$ 7.4 earthquake resulted from the further southeast migration of the strain energy of the 2001 Kunlun Mountains $M_S$ 8.1 earthquake along the EKFZ [24].

*5.2. Seismogenic Potential of the XDS of the EKFZ*

In the introduction, a detailed description was given of the scholars' analysis and study of various seismic data [7–9], model method [10], and coulomb stress variation [92–95], all of which indicate that with the occurrence of strong earthquakes in the surrounding area of the XDS of EKFZ, the possibility of strong earthquakes in this area increases significantly. Therefore, it is necessary to study the seismic potential of the XDS of the EKFZ.

The interseismic fault-locking technology has been widely used in focal fault risk assessment and earthquake potential magnitude estimation. During interseismic periods, the strengthening of fault contraction indicates that the coupling degree between fault planes in this region is high, increasing strain accumulation. Therefore, it can be considered that there is a high-incidence area of earthquakes in this region [19,30,96]. The interseismic locking area is constrained using the high precision and high-density geodetic observation data, and the approximate range of coseismic slip is determined based on it. Based on the long-term fault slip rate and the time of the recent earthquake, we can estimate accurately the potential magnitude of the earthquake in this study area. According to the distribution of the slip-deficit rate shown in Figure 8c, the moment accumulation ($M_0$) can be calculated through the first-order relation [29,30]. The relationship (Formula (2)) is as follows:

$$M_0 = \mu U A \tag{2}$$

where $A$ represents the area of the fault patch, $\mu$ represents the shear modulus, set to be 30 GPa, and $U$ represents the slip-deficit rate from the inversion. The XDS of EKFZ is a large seismic void region, but through geological mapping and dating, scholars have revealed that there have been at least six paleoseismic events in this section since the Holocene, and the last paleoseismic event was $1540 \pm 92$ aB.P. [3,7,15,97,98], so we surmised that the stress accumulation in the XDS of EKFZ lasted about $483 \pm 92$ years. Over 483 years, the cumulative elastic strain energy of the XDS of the EKFZ can produce an $M_W$ $7.29 \pm 0.1$ earthquake in the future. However, by using quantitative geological data and other methods, the former people speculated that there is an $M$ (7.5~8.0) earthquake risk in the XDS of the EKFZ [3,9,15,81], which is different from the research results in this paper. According to the literature review, the reasons may be as follows: (1) This paper does not consider the stress loading effect of the left-lateral slip of the 2001 Kunlun Mountains $M_S$ 8.1 earthquake on the XDS of EKFZ [7,92,93,97]; (2) The time of earthquake recurrence may

be inconsistently defined due to faulting habits and the incompleteness of paleoseismic events [3,9,15,81], and the time definition in this paper is small; (3) Combined with the statistical slip rate in Table 3, it can be inferred that the slip rate of InSAR inversion in this paper is smaller than that of geological estimation [15,99,100]. To sum up, the XDS is undoubtedly the most dangerous area for future large earthquakes in the EKFZ, which needs special attention.

**Table 3.** The slip rate from XDS to Xiugou Basin of EKFZ.

| Position | Dislocation Maker | Method | Age (ka) | Dislocation Distance (m) | Slip Rate (mm/yr) | Reference |
|---|---|---|---|---|---|---|
| XDS | Terrace dislocation | Cosmogenic nuclide | 2.4~11.8 | 23~174 | 10-15 | [99,101] |
| XDS | Terrace dislocation | Cosmogenic nuclide | 1.8~8.1 | 24~110 | 11.7 ± 1.5 | [15,81,98] |
| XDS | Terrace dislocation | Thermo luminescence | 262~335 | 2970 ± 30 | 11.6 ± 0.9 | [100] |
| XDS | NA | InSAR | NA | NA | 9.8 ± 2.3 | This study |
| Xiugou west | Alluvial fan dislocation | Thermo luminescence | 297 ± 19 | 2970 ± 30 | 10.1 ± 0.8 | [100,102] |
| Xiugou middle | Terrace dislocation | Cosmogenic nuclide | 6.3~8.1 | 90 ± 5 | 12.9 ± 2.9 | [15,81,98,100] |
| Xiugou | NA | InSAR | NA | NA | 8 ± 3 | This study |

*5.3. Variations of the Present-Day Slip Rate in the XDS of the EKFZ*

In this paper, there are some errors in the data collection and processing of the ascending and descending orbit. Therefore, to ensure the accuracy of data processing in advance, based on the discrete GPS data and the InSAR deformation rate field of the ascending and descending orbits, we obtained the three-dimensional crustal deformation field in the study area (Figure 9a–c), and the fault movement characteristics in the study area were obtained by inversion fitting. It can be seen that most regions have been uplifted, but the localization of this phenomenon is remarkable from Figure 9c, indicating that the vertical motion may contain a large number, including non-tectonic deformation. Therefore, we only considered the horizontal motion components and ignored the vertical motion components in this study. Since the InSAR deformation measurement is based on the reference range of radar LOS, it should first be converted into the deformation rate in the parallel fault direction through Formula (3), and the profile location was selected, as shown in Figure 9. The fault strike parameters at the profile were set to 90°, 88.95°, 94.96°, 104°, 97.20°, 100.5°, and 105.62°, which were all perpendicular to the fault strike. Secondly, according to Savage et al., the theoretical model formula of spiral dislocation (Formula (4)) was given to invert the current active motion characteristics of the study area [58]. The final fitting results in the parallel direction of the fault are shown in Figure 10.

$$D_{fault} = V_n \cos \beta + V_e \sin \beta \tag{3}$$

$$V_0 = V_{ref} + V * a \tan((X_0 - X)/D)/PI \tag{4}$$

where in Formula (2), $D_{fault}$ represents the parallel deformation rate, $V_n$ represents the north deformation rate, $V_e$ represents the east deformation rate, and $\beta$ represents the fault strike. In Formula (3), $V_0$ represents the observed value, $V_{ref}$ represents the overall offset, $V$ represents the fitting slip rate, $X_0$ represents the distance between the observed value and the fault, $X$ represents the distance between the fitting value and the fault, and $PI$ is π.

In this study, by using the discrete GPS data and the InSAR deformation rate field, the three-dimensional crustal deformation field of the XDS of the EKFZ was obtained. Using seven profiles of the InSAR deformations as constraints, the spiral dislocation model used estimated the locking depth of the XDS to be 29 ± 6 m, the average slip rate of the XDS (P1–P6) to be 9.8 ± 2.3 mm/yr, the locking depth of Xiugou Basin to be 26 m, and the average slip rate of Xiugou Basin (P7) slip to be 8 ± 3 mm/yr, among which the fitted locking

depth is basically consistent with the previous modeling results (Figure 8). The results of this paper are compared with the slip rate obtained by previous geological mapping and dating (Table 3, e.g., [15,99,100]). The average slip rate of the XDS to Xiugou Basin of EKFZ obtained by this study is basically consistent with them and is also close to the 10~12 mm/yr slip rate of the EKFZ observed by GPS today [20,86]. It can be seen from Figure 10 that the slip rate of the XDS gradually increases from west to east, and the fitting values of P1, P2, and P3 are relatively small, which may be caused by the sparse observed number of the InSAR deformation at these three profile locations, but it does not affect the accuracy of the results.

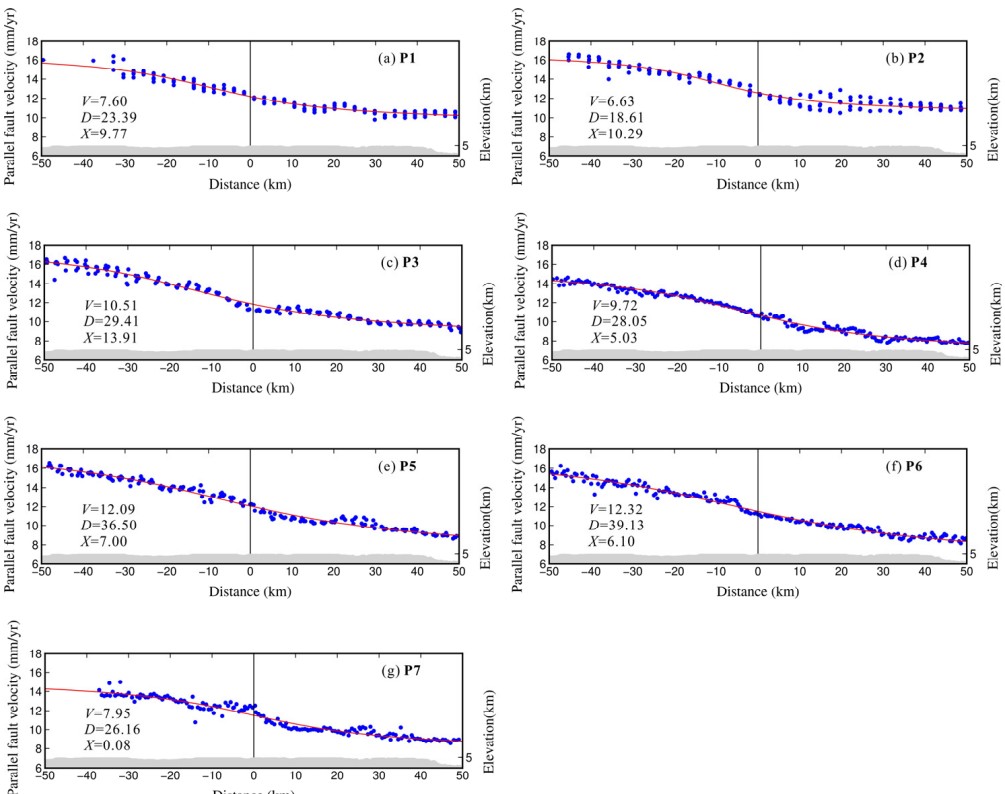

**Figure 10.** Section view of the XDS of the EKFZ along the parallel direction in Figure 9. The blue dots represent InSAR observations, the red line segments represent the best-fit values, the black vertical lines represent the location of the EKF (same as in Figure 5), and the gray areas represent topographic information (same as in Figure 5). The best fitting parameters are marked in the lower left corner of (**a–g**). *V* represents the modeled slip rate at depth (mm/year), *D* represents locking depth (km), and *X* represents the distance from fault location (km).

## 6. Conclusions

In this study, InSAR technology was used to obtain the InSAR interseismic deformation field of the XDS of the EKFZ based on the Sentinel-1 SAR data of the ascending and descending orbits, and the deformations were in good agreement with the GPS observations. Based on these constraints, the slip rate and locking depth of the region were derived by combining the triangular dislocation model and the spiral dislocation theoretical model, and the seismic potential of the XDS of the EKFZ was estimated.

The following conclusions could be drawn from this study: (1) The LOS deformation field of the XDS of the EKFZ was obtained by using Sentinel-1 data of ascending and descending orbits, which indicated that the XDS of the EKFZ is dominated by horizontal movement. Combined with the interference results, it is shown that the strike-slip rate dominates the deformation information of the XDS of the EKFZ. The deep strike-slip rate of the fault is about 6 mm/yr, the deep dip-slip rate is about 2 mm/yr, and the slip-deficit rate on the fault surface is about 6 mm/yr; (2) Combined with the spiral dislocation theory

model, the slip rate of the XDS to Xiugou Basin of the EKFZ has a gradually increasing trend, with an average slip rate of 9.6 ± 2.3 mm/yr and a locking depth of 29 ± 5 m; (3) The stress accumulation is about 483 ± 92 years in the XDS of EKFZ and indicates that the cumulative elastic strain energy of the XDS of the EKFZ can produce an $M_W$ 7.29 ± 0.1 earthquake in the future, which needs special attention.

**Author Contributions:** All the authors participated in editing and reviewing the manuscript. Conceptualization, L.J. and L.Z.; methodology, S.K. and N.L.; software, S.K. and J.X.; validation, F.J.; formal analysis, S.K. and L.J.; investigation, S.K. and L.Z.; resources, S.K. and J.X.; data curation C.L. and W.Z.; writing—original draft preparation, S.K.; writing—review and editing, S.K., L.J., L.Z., C.L. and W.Z.; visualization, S.K. and W.Z.; supervision, L.J. and N.L.; project administration, S.K., C.L. and J.X.; funding acquisition, S.K. and J.X. All authors have read and agreed to the published version of the manuscript.

**Funding:** This research was funded by the Natural Science Foundation of Shaanxi Province, China (Nos. 2023-JC-QN-0329, 2023-JC-QN-0292, and 2023-JC-QN-0296), the National Natural Science Foundation of China (No. 42104061), and the Spark Programs of Earthquake Sciences granted by the China Earthquake Administration (No. XH23059YA).

**Data Availability Statement:** The Sentinel-1 data used in this study are downloaded from the European Space Agency (ESA) through the ASF Data Hub website https://vertex.daac.asf.alaska.edu/.

**Conflicts of Interest:** The authors declare no conflict of interest.

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
