# Peer review of "Study of the Interseismic Deformation and Locking Depth along the Xidatan–Dongdatan Segment of the East Kunlun Fault Zone, Northeast Qinghai–Tibet Plateau, Based on Sentinel-1 Interferometry"

_remotesensing, doi:10.3390/rs15194666_

Round 1

Reviewer 1 Report

General comments:

The authors of this manuscript utilize Sentinel-1 InSAR data to analyze locking parameters for the Xidatan-Dongdatan segment of the East Kunlun fault zone. The scientific idea’s great but I have several major concerns that the I hope the authors could consider addressing before taking this manuscript to publication. 1) the logic of incorporating InSAR data for strain rate analysis is a bit unclear in the manuscript; 2) some improvements on the writings are needed; 3) choices of baselines and acquisitions are confusing; 4) assumptions of the underlying theory are not properly discussed; etc… Please see my detailed comments below. Overall, a major revision is required.

Detailed comments:

Title: seismogenic  locking, member  segment, images data  interferometry

Line 12: faults  fault systems

Line 12-12: “and is accompanied by left-lateral strike-slip characteristics in mainland 12 China. ”  and is characterized by left-lateral strike-slip.

Line 13: Xidatan-Dongdatan member (XDM)  Xidatan-Dongdatan segment (XDS)

Line 14: strain loading  stress loading

Line 15: “so this segment is widely regarded as”  so it is widely regarded as

Line 16: delete “images”, just Sentinel-1 data.

Line 20: delete “by”

Line 22: “movement”  motion

Line 22: “interference results”?

Line 26: “deep strike-slip rate”? I am not sure whether there is a deep version of slip rate. I think you mean the long-term slip rate.

Line 27: “slip loss”?

Line 29-30: You don’t need to repeat the previous sentence.

Line 45: seismic activity capacity? What is this supposed to mean?

Line 54-58: this is basically repeated in the abstract.

Line 60: a variety of basic data such as “from” seismic geology, geodesy, … geology/geodesy/geophysics are not data.

Figure 1: mark the rough rupture area on the map for the past events.

Line 68: Maqu member  Maqu segment

Line 71-72: “the future earthquake form is more severe”. An earthquake form is severe?

Line 85-86: Put year and magnitude before the name of an earthquake.

Line 87-88: What is “practical significance”?

Line 88: “lock-up” degree? Coupling or locking?

Line 92: “we can not only”  “not only can we ”

Line 97: I would not call the coverage time being the same if they have about two year’s difference while the total being five.

Line 98: peculiarities? Do you mean “asperities”?

Line 99: “combined with InSAR technology”, does this go with the previous or next sentence?

Line 101-102: “the urgency of earthquake generation in this area is analyzed”. You analyzed the urgency? What’s this sentence supposed to mean?

Just reading from the abstract and intro, I would like to recommend the authors to consider rephrasing lots of their descriptions in the manuscript, or find more proper expression on some of the scientific issues. The wording choices are a really uncommon to me. Since this is a systematic issue throughout the manuscript, I’ll make no further comment on this later on.

Line 139: I think 120x30 multi-looking for Sentinel-1 TOPS data are about 300m x 400m rather than 450m x 450m.

If you selected 63 out of 64 acquisitions for ascending track 173, how come in figure 3a, there’s no acquisition number 1 and 2? Also, what’s the criterion for selecting acquisitions? To me, acquisition NO.4 looks like a good choice. Why did you exclude it from the stack? Also put the number of remaining acquisitions in table 1 as well.

Line 148-151: This is really confusing to me, how come that one has to set perpendicular baseline larger than a specific number, considering the effect to data quality from increased perpendicular baseline is monotonic. Please explain in the manuscript.

Line 183-185: This estimate requires your target fault being the only fault that’s accumulating moment in your observation area. Is that really the case? What are the gray lines in Figure 4?

Line 192: ESD can only help azimuthal coregistration.

Line 196: I think this is called multi-looking.

Line 198-199: If you have 8/2 multi-looking and then a 120/30 multi-looking, the resulting solution will be 960/60. Explain in the context.

Line 201-203: I think GACOS not only corrects stratified component of atmosphere, but also some of the turbulent part as well.

Line 203-206: So basically, you only keep quite limited info from InSAR, since the final product only contains temporally low-passed and spatially high-passed component. What if you only use GNSS to do the estimates on slip rates? Any differences? If you didn’t control the long wavelength signal with GNSS, after such steps, how could you ensure the accuracy over long distances?

Line 236-237: How could you compare velocities from different orbits considering they have different look angles?

Line 239-240: Why naming AA’ and DD’, where’s BB’ and CC’? Also connect AA’ and DD’ in the plot.

Line 280: Any chance that the dipping angle is larger than 90? i.e. some degrees dipping to the other side?

Figure 8 is too saturated to see any details at depth. Consider using a more stretched color scale.

Section 5.1: How are the strain rates produced? This covers a much larger area than your InSAR data coverage. If you used GNSS data only, what’s the purpose of this study?

Line 401: U instead of mu.

Line 407: do not repeat the same sentence. Also what equation 2 gives you is the moment, to convert it to magnitude of an earthquake, you need to take a logarithmic function. How come a linear +/- 92 years result in a linear +/- 0.1 magnitude after taking a log10? This does not make sense and the authors should explain in the manuscript.

Line 421-422: Usually for a strike slip system like EKFZ, one should ignore the normal component and estimate the fault parallel and vertical components.

See above

Reviewer 2 Report

This paper collects Sentinel-1 image data of the Xidatan-Dongdatan member (XDM) in the East Kunlun fault zone (EKFZ) from July 2014 to July 2019. The slip rate and locking depth of the Xidatan-Dongdatan member (XDM) in the East Kunlun fault zone (EKFZ) were obtained by using the Interferometric Synthetic Aperture Radar (InSAR) technique to obtain the interseismic deformation field, and the seismic potential of this segment was analyzed. From the perspective of the results, the analysis results proposed in this paper have certain academic reference significance, but the prediction of earthquakes should be extremely rigorous, because this may cause public panic. This requires the author to consider and consider this part of the language description carefully, and there are some problems in the text that need to be answered or modified by the author :

1. In the abstract part, the author has the following contents that need to be answered or modified by the author :

a) It is not possible to see any logical connection to the timing of the selection of the image data from the context of the authors' abstract, and the authors are asked to explain why they chose the image data from this period of time and not others, and to revise the abstract description appropriately;

b) Conclusion three of the author's abstract, regarding the accumulation of seismic stress and thus the prediction of a possible earthquake of about Mw 7.29 in the fault zone, asks the author to be more careful about this point, which should be taken with extreme rigor regarding earthquake prediction.

2. In the first part, the author has the following contents that need to be answered or modified by the author :

a) The focus of the introduction should be primarily on the subsequent development of the study, and it should make clear the relevant links between the context of the subsequent study and the study's content;

b) The authors are still unable to see any logical connection to the timing of the selection of the image data in their description in the introductory section.

3. In the second part, the author has the following contents that need to be answered or modified by the author :

a) In this section, the cyan label in Figure 3 does not show up well, and it is recommended that the authors replace it with another color;

b) The paper mentions " and a reliable and more informative constrained least square method is used 183 to invert the interseismic slip distribution of the XDM of EKFZ ", can the authors give the specific experimental procedure of this constrained least squares method in this paper.

4. In the third part, the author has the following contents that need to be answered or modified by the author :

a) There is a great deal of language in the author's description in section 3.1 that duplicates the previous description, and the author is asked to change it;

b) The author describes in 3.1 that the GPS spatial density in the region is seriously insufficient, and the monitoring accuracy is significantly reduced, and then describes in part 3.2 that the accuracy of the InSAR velocity field is determined by the GPS results, and at the same time, the GPS point locations only appear as 5 stations in the accuracy evaluation of this paper, which seems to be insufficient and affects the persuasive power of this part, and it is suggested that the author changes the accuracy evaluation again after careful deliberation.

5. In the fourth part, the author has the following contents that need to be answered or modified by the author :

a) In section 4.1, the authors set the fault dip of the Xidatan-Dongdatan member (XDM) in the East Kunlun fault zone (EKFZ) to 90°, with 90° as the critical degree of dip. The authors are requested to carefully consider whether there is a problem with the search algorithm used, which is stuck in an algorithmic loop after searching for the critical degree, and to explain this accordingly;

6. In the discussion section, the timing of the authors' image data selection does not seem to be sufficient to support the study of strain accumulation and other related changes in the Xidatan-Dongdatan member (XDM) of the East Kunlun fault zone (EKFZ) after July 2019. After 2019, there have been other large earthquakes in its immediate environment, such as the Maduo earthquake mentioned by the authors. That is, the timeliness of the article is insufficient.

7. There are a large number of grammatical errors in this paper, and the authors are advised to check and carefully change the entire text.

The authors are requested to answer the above questions and to scrutinize the entire text and make corrections based on the suggestions made above, as well as to refine the articles and statements.

There are a large number of grammatical errors in this paper, and the authors are advised to check and carefully change the entire text.

Round 2

Reviewer 1 Report

I don't have any further comments, but some of my concerns did not get fully addressed I believe. e.g., the authors put together a criterion to select interferograms but does not properly describe the reason behind it; a single fault assumption made while multiple faults exists in the region (symmetry only tells you which one is the main fault, not necessarily other faults don't matter); more dipping angle options were not tested (though you cited previous work). Though these maynot fully overturn the conclusions from the paper, they could be buggy to read. I sincerely hope the authors could consider the above concerns before publishing their work. 

Needs some proofreading. 

Author Response

请参阅附件。
